# Factor XI in Carriers of Antiphospholipid Antibodies: Elevated Levels Associated with Symptomatic Thrombotic Cases, While Low Levels Linked to Asymptomatic Cases

**DOI:** 10.3390/ijms242216270

**Published:** 2023-11-13

**Authors:** Javier Pagán-Escribano, Javier Corral, Antonia Miñano, José Padilla, Vanessa Roldán, María Julia Hernández-Vidal, Jesús Lozano, Isabel de la Morena-Barrio, Vicente Vicente, María Luisa Lozano, María Teresa Herranz, María Eugenia de la Morena-Barrio

**Affiliations:** 1Servicio de Medicina Interna, Unidad de Enfermedad Tromboembólica, Hospital General Universitario José María Morales Meseguer, 30008 Murcia, Spain; pagan02468@gmail.com (J.P.-E.); majirulihv@gmail.com (M.J.H.-V.); jesuslozanoh@gmail.com (J.L.); 2Servicio de Hematología Hospital General Universitario José María Morales Meseguer, Centro Regional de Hemodonación, Universidad de Murcia, IMIB-Pascual Parrilla, CIBERER-ISCIII, CEI Campus Mare Nostrum, 30003 Murcia, Spain; javiercorraldelacalle@gmail.com (J.C.); antoniadpminano@gmail.com (A.M.); josepadillaruizcrh@gmail.com (J.P.); vroldans@um.es (V.R.); vicente.vicente@carm.es (V.V.); mllozano@um.es (M.L.L.); 3Servicio de Reumatología, Hospital Clínico, 46010 Valencia, Spain; delaeme84@gmail.com

**Keywords:** factor XI, coagulation, antiphospholipid syndrome, antiphospholipid antibodies, thromboembolic disease

## Abstract

Antiphospholipid syndrome (APS) is a thromboinflammatory disorder caused by circulating antiphospholipid autoantibodies (aPL) and characterized by an increased risk of thrombotic events. The pathogenic mechanisms of these antibodies are complex and not fully understood, but disturbances in coagulation and fibrinolysis have been proposed to contribute to the thrombophilic state. This study aims to evaluate the role of an emerging hemostatic molecule, FXI, in the thrombotic risk of patients with aPL. Cross-sectional and observational study of 194 consecutive and unrelated cases with aPL recruited in a single center: 82 asymptomatic (AaPL) and 112 with primary antiphospholipid syndrome (APS). Clinical and epidemiological variables were collected. The profile of aPL was determined. Plasma FXI was evaluated by Western blotting and two coagulation assays (FXI:C). In cases with low FXI, molecular analysis of the F11 gene was performed. FXI:C levels were significantly higher in patients with APS than in patients with AaPL (122.8 ± 33.4 vs. 104.5 ± 27.5; *p* < 0.001). Multivariate analysis showed a significant association between symptomatic patients with aPL (APS) and high FXI (>150%) (OR = 11.57; 95% CI: 1.47–90.96; *p* = 0.020). In contrast, low FXI (<70%), mostly caused by inhibitors, was less frequent in the group of patients with APS compared to AaPL (OR = 0.17; 95%CI: 0.36–0.86; *p* = 0.032). This study suggests that FXI levels may play a causal role in the prothrombotic state induced by aPLs and holds the promise of complementary treatments in APS patients by targeting FXI.

## 1. Introduction

Antiphospholipid syndrome (APS) is a systemic autoimmune disease in which multiple and distinct procoagulant and proinflammatory factors, triggered by the presence of high and persistent levels of antiphospholipid antibodies (aPL), interact to cause thromboembolic events and/or obstetric complications [1,2]. Although several risk factors and markers have been identified to be involved in the risk of thrombotic events in patients with aPL, the identification of new prognostic biomarkers is still a major challenge. In this context, clotting factor XI (FXI) is an excellent candidate for consideration as a potential contributor to the risk of thrombotic events.

FXI is a member of the contact pathway, which bridges the initiation and amplification phases of coagulation [3]. There is evidence that aPLs interact directly with FXI [4,5]. Under physiological conditions, β-2-glycoprotein-1 (β2GP1) attenuates the intrinsic coagulation pathway by inhibiting thrombin activation of FXI [4], and anti-β2GP1 antibodies (aβ2GP1) potentiate the in vitro inhibition of β2GP1 on FXI [5]. Variations in FXI may have relevant hemostatic consequences. High FXI levels (>150% of reference values) are associated with increased thrombotic risk [6]. In contrast, FXI deficiency (<70%), a rare coagulopathy with mild bleeding consequences [7,8], has strong antithrombotic protection [9,10], making FXI a promising target for antithrombotic therapy [11].

aPL carriers who do not have a history of thrombosis are not routinely prescribed anticoagulant prophylaxis unless specific thrombotic risk factors outweigh the risk of major bleeding. The ability to stratify aPL carriers into high and low thrombosis risk groups would be of great value, allowing clinicians to intervene with prophylaxis when indicated. There is increasing interest in evaluating the contribution of concomitant thrombophilic risk factors to the presence of aPL for venous or arterial thrombotic events. In aPL carriers, we report here the association of FXI levels with the presence of previous thrombotic events.

## 2. Results

### 2.1. Patient Characteristics, Thrombotic, and Antiphospholipid Profile

For the study, 194 consecutive and unrelated cases aged ≥18 years with aPL were recruited: 82 with AaPL and 112 with primary APS. The clinical and demographic characteristics of all enrolled participants are shown in Table 1. Males were overrepresented in the APS group compared to the AaPL group. In addition, hypertension, dyslipidemia, and obesity (BMI > 30 kg/m^2^) were more common in the APS group than in the AaPL group (Table 1). Overall, 11.0% of AaPL patients had no cardiovascular risk factor (CVRF) compared to 2.8% of APS cases. According to the Systematic Coronary Risk Evaluation (SCORE) model, a significantly higher number of patients in the APS than in the AaPL groups had a moderate SCORE (42.0% vs. 24.4%, *p* = 0.014) (Table 1).

Venous thrombosis was the most common initial vascular event in the APS group (73.3% of all initial thromboses). Deep vein thrombosis accounted for 49.5% of all initial thrombotic events. Non-cardioembolic stroke was the most common arterial event, accounting for 19% of all initial thrombotic events. Thrombotic recurrence was observed in 37 of the patients with primary thrombotic APS (35.2%), with a mean follow-up of 91 months between the first thrombotic event and recurrence. Twenty-three of thirty-seven (62.2%) patients with recurrent APS were not receiving any type of antithrombotic treatment; 5 (13.5%) were receiving antiplatelet therapy.

Regarding the use of anticoagulant therapy, no subjects with DVT were on anticoagulant therapy prior to the first thrombotic event. In the APS group, 90 (85.7%) patients were receiving anticoagulant treatment with anti-vitamin K at enrollment.

The profile of aPL in the APS and AaPL groups is shown in Table 2. aβ2GP1 antibodies were more frequent in the AaPL group. The thrombotic risk of each group according to the aPL profile was not significantly different. 

### 2.2. Factor XI Coagulant Activity

FXI:C was determined at a median of 5 years (IQR: 9) after the first thrombotic event in APS patients. In this group, the median time on anticoagulant treatment from the last thrombotic event to the determination of FXI:C was 48 months (IQR 60).

As previously described [12], FXI:C levels determined with silica do not have a normal distribution and yield lower values than those obtained with ellagic acid in the entire cohort of subjects included in this study (APS, THROMB, AaPL, and HC groups): 105.3% (IQR 36.8) and 114.3 ± 31.3%, respectively (*p* = 0.001).

FXI:C levels were statistically higher in APS patients than in AaPL or HC (*p* < 0.05 for all comparisons), regardless of the contact activator used (Figure 1A). Considering the clinical consequences that elevated or low FXI:C may have, we evaluated the proportion of cases with high FXI:C (>150%) or low FXI:C (<70%) in the groups of subjects included in our study. We performed this analysis with the FXI:C values obtained with each activator individually and also when the results were consistent with both techniques (global result).

As shown in Figure 1B, the percentage of patients with FXI:C > 150% was significantly higher in the APS and THROMB groups than that found in subjects without thrombosis, including AaPL and HC. Interestingly, FXI:C deficiency was identified mainly in the AaPL group (Figure 1C).

Univariate analysis of FXI:C in the two groups of subjects with aPL showed significantly higher levels in the APS group than in the AaPL group (Table 3). This is consistent with both the higher prevalence of cases with high FXI:C (>150%) in APS and with low FXI:C (<70%) in AaPL. Thus, the detection of FXI:C >150% by both methods in an individual with aPL was associated with a 14.95-fold higher incidence of thrombosis (APS patients) (Table 3). Interestingly, the only AaPL subject with FXI:C > 150% in both tests developed major venous thromboembolism in October 2022. This is a 57-year-old woman with no other risk factors for venous thrombosis who is triple positive for aPL (AL and IgG isotype of aβ2GP1 and aCL). In contrast, FXI deficiency (FXI:C < 70%) was overrepresented (4.93-fold) in AaPL patients compared to APS patients (Table 3). 

This analysis was performed with the FXI:C values obtained with each activator: (1) SynthASilTM (SS) (colloidal silica-based reagent) and (2) SynthAFaxTM (SFX) (ellagic acid-based reagent), individually, and also when the results were consistent with both techniques (Global result).

There was no correlation between FXI:C levels and the specificity or profile of APS in the entire cohort of APS patients (Appendix A). However, there was a moderate and negative correlation between the FXI coagulation activity measured with silica (SynthASilTM) and aβ2GP1 and aCL antibody titers for both isoforms, IgM, and IgG (Appendix A). When ellagic acid (SynthAFaxTM) was used to determine FXI:C, the negative correlation with aβ2GP1 levels was maintained (Appendix A).

Western blot analysis of samples from cases with FXI:C >150% confirmed an increase in plasma FXI levels (Figure 2A). However, results were conflicting when the 14 cases with FXI:C < 70% (10 AaPL; 3 APS; 2 THROMB and 1 HC) were evaluated by Western blotting, as only four cases had a confirmed significant deficiency of FXI antigen in plasma, similar to that found in the CONGFXIDEF group (Figure 2B).

Molecular analysis of *F11* in all 16 subjects with FXI:C deficiency (<70%) identified only heterozygous pathogenic variants in the four cases that not only had reduced FXI:C but also confirmed FXI antigen deficiency in plasma (Appendix A). Two unrelated patients with aPL (one APS and one AaPL) shared the same variant: c.403G > T; p.Glu135Ter, a recurrent mutation in the Ashkenazi population causing deficiency with reduced plasma antigen levels [13]. A second AaPL case carried two contiguous *F11* defects, a synonymous c.801A > G, p.Thr267Thr, and a missense variant c.802C > T, p.Arg268Cys. The p.Arg268Cys mutation has also been described in the FXI mutation database https://www.factorxi.org/ (accesed on 29 September 2023) and HGMD (CM035499) and has been identified in four patients with FXI deficiency [14,15]. In addition, a healthy blood donor carried the c.1327C > T, p.Arg443Cys and HGMD (CM062624), which was identified previously in two patients with congenital FXI deficiency [14,16]. Family studies confirmed the segregation of the genetic defects identified in this study with FXI deficiency n.

In the remaining aPL-positive cases with FXI:C < 70% but normal FXI levels by Western blotting, no genetic defect in *F11* was detected by either sequencing or MLPA analysis. In these cases, the presence of an FXI inhibitor was confirmed by dilution assays.

In multivariate analysis, the factors that were independently overrepresented in patients with APS compared to AaPL were AL (OR 2.22), (OR 2.16), aβ2GP1 IgG (OR 0.38), FXI:C levels > 150% (OR 11.57), and FXI:C < 70% (OR 0.18) (Table 4).

## 3. Discussion

This study suggests that circulating FXI may modify the properties of aPL and modulate the associated prothrombotic state. In fact, FXI:C > 150% was the factor with the strongest association with aPL-associated thrombosis (OR 11.57), while that of LA and dyslipidemia was only around 2-fold. Unlike most thrombotic diseases, which are caused by occlusion of either the venous or arterial circulation alone, aPL antibodies can induce a venous and/or arterial thrombotic event. Previous studies have shown that elevated levels of FXI are a known risk factor for thrombosis in the general population, which may be associated with an increased risk of both venous and arterial thrombosis [17,18]. Preliminary data also suggest that APS patients have higher than expected circulating levels of the active free thiol form of factor XI [19]. 

The contact system is a simple pathway consisting of four elements, FXII, FXI kallikrein, and high molecular weight kininogen. However, it is an intersection of three important systems: coagulation, inflammation, and the immune response [20]. Over the past decade, in-depth studies of this pathway have provided the basis for exciting opportunities to uncover information about the pathogenesis of thrombotic and inflammatory disorders [20,21]. However, perhaps the most important paradigm shift regarding this pathway has been the antithrombotic protection associated with deficiency or inhibition of all elements of the contact pathway, but primarily FXI. This has been demonstrated in animal models and epidemiological studies [10,22,23]. These findings have stimulated the development of drugs targeting the elements of the contact pathway, mainly FXI, using various approaches to provide safer thromboprotection than current antithrombotic treatments [24,25,26,27,28].

APS is a serious thrombotic disorder triggered by aPLs in combination with additional factors, such as CVRF [1]. Nevertheless, further prognostic markers are needed to dissect under what circumstances patients with aPL develop thrombotic complications [29]. In this context, considering that FXI is a phospholipid-binding protein [30] and the relationship between β2GP1 and FXI [4,5], it seems appropriate to evaluate the role of FXI levels in the development of thrombosis in patients with aPL.

To date, only one study has suggested a role for FXI in APS. Giannakopoulos et al. found increased levels of reduced Cys362-Cys482 and Cys118-Cys147 disulfide bonds in FXI in a small and heterogeneous sample of 10 APS patients compared to 20 patients without aPL or thrombotic complications and 15 AaPL patients. Since reduced FXI showed increased activation by thrombin, FXIIa, or FXIa compared to non-reduced FXI, these authors suggested that this mechanism may contribute to the venous and arterial thrombosis seen in APS patients [19]. The present study is the first to demonstrate an independent role for FXI levels in patients with primary APS and AaPL, supporting the dual role or antagonistic effect of this contact element in modulating thrombotic risk. High FXI levels (>150%) would be associated with thrombotic risk, as they are 11-fold higher in patients with APS compared to patients with aPL. Conversely, low FXI levels (<70%) may protect against thrombosis, as low FXI levels were four times more common in patients with positive aPL than in those who developed APS. Although cases of congenital FXI deficiency have been identified, in our cohort, FXI deficiency was mainly acquired, probably caused by aPL itself. Thus, neutralization of FXI by aPL may reduce or abolish its prothrombotic consequences and protect against a thrombotic event, mimicking the effect of therapeutic anti-FXI antibodies. Consistent with previous studies, we found an inversely proportional correlation between aβ2GP1 levels and FXI coagulation activity [4,5]. The overrepresentation of AaPL vs. APS in this group of patients might be explained by the protective role of reduced FXI activity.

Current antithrombotic therapy in patients with APS is unsatisfactory due to a high rate of thrombotic recurrence [31]. Several lines of evidence suggest that FXI may be associated with inflammation. A significant number of proteins involved in thrombo-inflammation are associated with FXI:C [32]. In addition, animal studies show that FXI deficiency attenuates inflammation through contact system and cytokine-regulated mechanisms [33] and that inhibitors of FXI activation reduce coagulopathic and inflammatory responses [34]. Selectively targeting elevated FXI levels in aPL carriers rather than downstream hemostatic enzymes may not only reduce the procoagulant state associated with the presence of these antibodies but may also be an attractive approach to limiting the proinflammatory state associated with this condition. Overall, the results of this study, together with the low incidence of spontaneous bleeding observed in patients with FXI deficiency [35] and also in patients treated with anti-FXI drugs [24,25,26,27,28], strongly encourage the evaluation of new FXI-targeted strategies in patients with APS.

## 4. Material and Methods

### 4.1. Study Design

This is an observational, cross-sectional, and descriptive study of patients identified as carriers of aPL at the Morales Meseguer University General Hospital, Spain. Patients with aPL were followed up annually at the Autoimmune and Systemic Diseases Clinic of the Department of Internal Medicine. In addition, electronic medical records were evaluated to identify possible thrombotic events or other complications in all carriers of aPL.

### 4.2. Study Subjects

We included consecutive and unrelated subjects ≥18 years of age with a positive aPL test, either asymptomatic carriers of aPL (AaPL) and/or patients diagnosed with primary thrombotic and/or obstetric APS according to the 2006 Sydney criteria [1]. Patients with APS associated with other connective tissue diseases or patients with antinuclear antibodies ≥1/160 and/or positive extractable nuclear antigen antibodies were excluded.

In addition, the following patient groups were included in the study: 

HC: Healthy blood donor controls, all of whom were negative for aPL. The HC group was increased to 222 subjects for the determination of FXI:C. 

THROMB: Patients were recruited from hospital databases with an image-based diagnosis of arterial and/or venous thromboembolic events, excluding patients with atrial fibrillation and those with positive aPL. The inclusion of the THROMB group allowed us to confirm the relevance that high FXI levels increase the risk of thrombosis, also in patients with aPL, and to support that deficiency of FXI, mainly caused by the aPL, is thromboprotective. No subject was receiving anticoagulant treatment prior to the first thrombotic event. 

CONGFXIDEF: Patients diagnosed with congenital FXI deficiency carrying the same recurrent and founder defect in *F11*: p.Cys56Arg. These subjects were identified in a previous study of our group and are used as a control group for FXI deficiency [36].

All patients were recruited from a single center. All included subjects gave informed consent to participate in the study; all procedures followed adhered to the ethical standards of the Institutional Research Committee and the Declaration of Helsinki of 1964 and its subsequent amendments. The study was approved by the Human Research Ethics Committee of the Morales Meseguer University General Hospital, Murcia, Spain.

### 4.3. Blood Collection and Plasma Preparation

Blood was collected in citrate tubes by puncture of the antecubital vein using a Vacutainer^®^ (Becton–Dickinson). The tubes were centrifuged immediately (less than 2 h after collection) at 2200 g for 20 min. Plasma samples were aliquoted and stored at −80 °C until analysis. DNA was purified from buffy coats by standard salt-out procedures and stored at −20 °C until genetic analysis. 

### 4.4. Determination of Antiphospholipid Antibodies

We evaluated the antibodies included in the APS analytical criteria: lupus anticoagulant (LA), anticardiolipin (aCL), and aβ2GP1.

LA was detected by combining two phospholipid-dependent coagulation tests: the diluted Russell’s Viper Venom test and an LA-sensitive aPTT test using silica as an activator and a low phospholipid concentration, the Silica Clotting Time (Instrumentation Laboratory Diagnostic, Werfen, Barcelona, Spain) [37]. 

Semi-quantitative methods (QUANTA Flash^®^ aCL microparticle chemiluminescent immunoassays; Werfen, Barcelona, Spain) were used for the determination of aCL (IgG and IgM). The results are expressed in chemiluminescent units (CU). The cut-off values of 95 and 31 CU for IgG and IgM isotypes, respectively, established by Lakos et al. [38], were used. 

A fully automated chemiluminescent immunoassay using the ACL AcuStar system (Instrumentation Laboratory Diagnostic, Werfen, Barcelona, Spain) was used to determine aβ2GP1 (IgM and IgG). The cut- off for a positive result was set at the 99th percentile, which is the normal range for citrated plasma samples from healthy donors. In our population, this corresponds to 20 U/mL.

Cases with LA or double (any combination of LA, aCL, or aβ2GP1 antibodies) or triple (all three subtypes) aPL positivity or persistently high aPL titers were considered to have a high-risk aPL profile. 

### 4.5. Coagulation Assays

All samples were thawed and analyzed on the same day. Coagulation assays were performed immediately after thawing, under identical conditions, using the same lot of reagents for all samples.

In our APS cohort, the mean time from the first thrombotic event to the determination of FXI at the time of enrollment was 7.7 ± 7.6 years.

FXI:C values were determined using FXI-deficient plasma in an automated coagulometer (ACL TOP 550CTS, Instrumentation Laboratory, Werfen) according to the manufacturer’s instructions. Two different reagents, both from HemosIL (Werfen), were used to activate the contact pathway: (1) SynthASilTM (colloidal silica-based reagent) and (2) SynthAFaxTM (ellagic acid-based reagent), using the manufacturer’s standards, calibrators and instructions [12]. Results for FXI:C were expressed as a percentage of a laboratory reference standard.

### 4.6. Analysis of Plasma FXI by Western Blotting

Plasma FXI was detected and semi-quantified by Western blotting using a specific goat anti-human FXI polyclonal antibody (GAFXI-AT 190R1, Enzyme Research Laboratories, Swansea, UK) as previously described [39]. 

### 4.7. Molecular Characterization of FXI Deficiency

In cases of FXI deficiency detected by the two FXI:C methods, *F11*, the gene encoding FXI, was evaluated as previously described [39]. Briefly, the entire *F11* gene was sequenced using PGM (Ion Torrent, Madrid, Spain). Pathogenic gene variants were verified by Sanger sequencing. Gross gene defects were screened by multiplex ligand-dependent probe amplification (MLPA ^®^, Holland, The Netherlands).

### 4.8. Statistical Analysis

IBM Statistical Package for Social Sciences (SPSS 23.0) software was used for statistical analysis. The normality of continuous variables was assessed by Kolmogorov–Smirnov test. Data are presented as mean ± standard deviation or median (P25-P75) for normal and non-normal continuous variables. Qualitative variables are expressed as percentages. Pearson’s chi-squared test and Fisher’s exact test or linear-by-linear derivations were used to compare proportions or ordinal variables. The t-test, analysis of variance (ANOVA), or nonparametric Mann–Whitney or Kruskal–Wallis U tests were used to determine associations between quantitative or continuous variables. For continuous quantitative variables, we used Pearson’s test for normally distributed variables and Spearman’s test for non-normally distributed variables. Various logistic regression models were used for multivariate analysis. Differences between the two groups were considered statistically significant when *p* < 0.05.

## 5. Conclusions

This study suggests that FXI levels may play a causal role in the prothrombotic state induced by aPLs and holds the promise of complementary treatments in APS patients by targeting FXI.

## Figures and Tables

**Figure 1 ijms-24-16270-f001:**
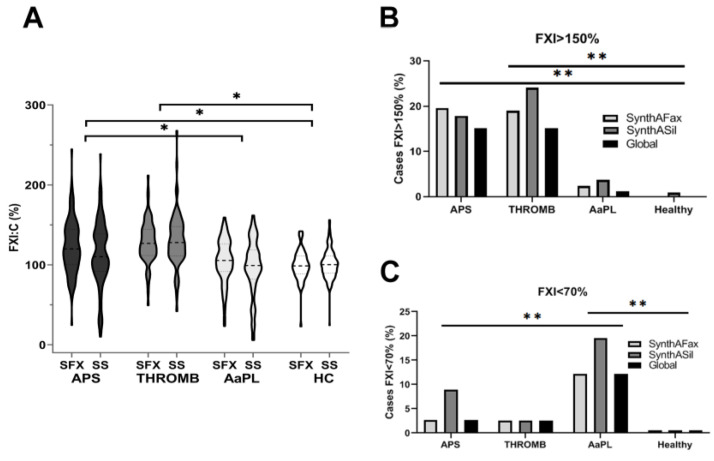
FXI levels in the groups of subjects included in the study: patients with antiphospholipid syndrome (APS); patients with thrombosis (THROMB); asymptomatic carriers of antiphospholipid antibodies (AaPL); and healthy controls (HC) (**A**) Violin plot of FXI:C values observed with the two contact activators, SynthAFax (SFX) and SynthASil (SS). * *p* < 0.05. (**B**) Percentage of subjects with high levels of FXI (>150%) observed by using SFX, SS or with both methods (global). ** *p* > 0.01. (**C**) Percentage of subjects with low levels of FXI (<70%) observed by using SFX, SS, or with both methods (global). ** *p* > 0.01.

**Figure 2 ijms-24-16270-f002:**
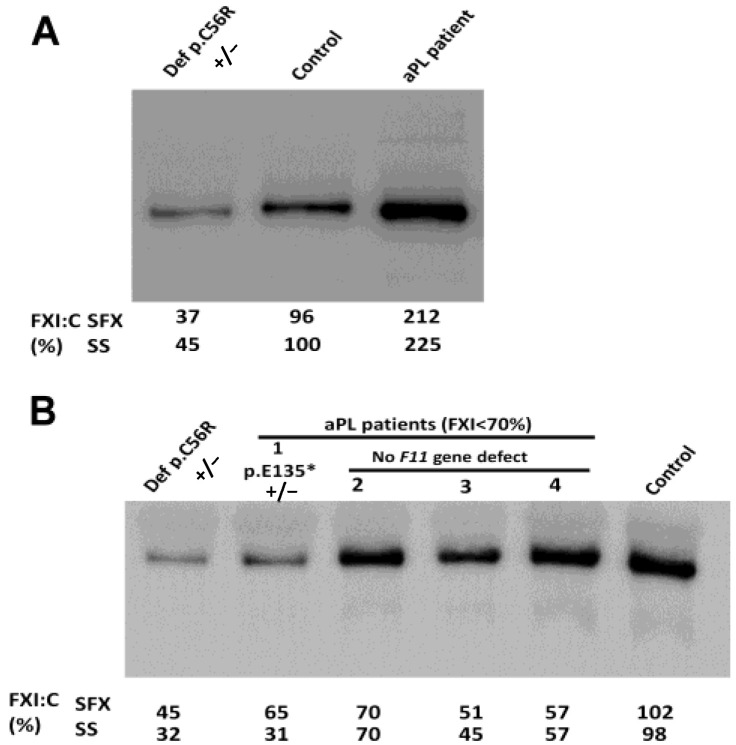
Identification of FXI in plasma by Western blotting in representative cases with high (**A**) and low (**B**) FXI:C. As controls, a pool of 100 healthy blood donors (control) and plasma from a patient with congenital FXI deficiency caused by the heterozygous p.Arg56Cys from the CONGFXIDEF group were also examined. The FXI:C values obtained with the two contact activators of these cases are also shown. SFX: SynthAFax^®^; SS: SynthASil^®^.

**Table 1 ijms-24-16270-t001:** Demographic and clinical characteristics of patients enrolled in this study.

	APS (n:112)	AaPL (n:82)	Univariate Analysis *p* (OR; 95%CI)	THROMB (n:79)	HC (n:74)
**Sex;** Male N (%)	58 (51.8)	28 (34.1)	**0.008 (2.23; 1.24–4.01)**	45 (57)	41 (55.4)
**Age-years;** Mean ± SD	51.3 ± 12.8	50.2 ± 15.1	0.586	51.9 ± 13.2	51.4 ± 11.5
**CVRF**					
Hypertension N (%)	50 (44.6)	25 (30.5)	**0.042 (1.87; 1.02–3.41)**	38 (48)	20 (27.0)
Dyslipidemia N (%)	84 (75)	43 (52.4)	**0.001 (2.72; 1.48–5.00)**	58 (73.4)	24 (32.4)
Diabetes Mellitus N (%)	11 (9.8)	9 (11.0)	0.794 (1.13; 0.45–2.87)	18 (22.8)	4 (5.4)
BMI > 30 Kg/m^2^ N (%)	43 (38.4)	17 (20.7)	**0.007 (2.45; 1.27–4.73)**	32 (40.5)	11 (14.9)
Smoking N (%)	66 (58.9)	44 (53.7)	0.465 (1.24; 0.69–2.20)	46 (58.2)	39 (52.7)
Sedentarism N (%)	57 (50.9)	49 (59.8)	0.247 (1.41; 0.79–2.51)	45 (57)	29 (39.2)
Any CVRF	110 (98.2)	73 (89.0)	**0.016 (6.78; 1.42–32.28)**	76 (96.2)	63 (85.1)
1 CVRF	17 (15.2)	16 (19.5)	0.428 (1.36; 0.64–2.87)	9 (11.4)	25 (33.8)
2 CVRF	29 (25.9)	21 (25.6)	0.964 (1.02; 0.53–1.95)	19 (24.1)	21 (28.4)
3 or more CVRF	64(57.1)	36 (43.9)	0.069 (1.70; 0.96–3.03)	48 (60.8)	17 (23.0)
**SCORE**					
Low SCORE N (%)	49 (43.8)	47 (57.3)	**0.043 (0.55; 0.30–0.98)**	ND	ND
Moderate SCORE N (%)	47 (42)	20 (24.4)	**0.014 (2.20; 1.17–4.14)**	ND	ND
High SCORE N (%)	10 (8.9)	7 (8.5)	0.957 (1.03; 0.37–2.83)	ND	ND
Very high SCORE N (%)	4 (3.6)	5 (6.1)	0.397 (0.56; 0.15–2.15)	ND	ND

We classified patients as “never smokers” if they had never smoked, “former smokers” if they had stopped smoking more than 6 months before the thrombotic event, and “current smokers” if they were active smokers at the time of the thrombotic event and/or had stopped smoking less than 6 months before the thrombosis, or at the inclusion in the study in patients without thrombosis. Abbreviations: Asymptomatic carriers of antiphospholipid antibodies (AaPL); Antithrombin (AT); Cardiovascular risk factors (CVRF); Deep venous thrombosis (DVT); Factor V Leiden (FVL); Healthy controls (HC); Primary antiphospholipid syndrome (APS); Standard deviation (SD); Patients with a history of thrombosis and with negative antiphospholipid antibodies (THROMB).

**Table 2 ijms-24-16270-t002:** Profile of antiphospholipid antibodies in patients carrying aPL, with (APS) or without (AaPL) vascular complications.

	APS (n:112)	AaPL (n:82)	Univariate Analysis *p* (OR; 95%CI)
aPL Profile			
LA N (%)	86 (75)	50 (61)	**0.023 (2.07; 1.11–3.86)**
aCL N (%)			
IgM	21 (18.8)	19 (23.2)	0.453 (0.77; 0.38–1.54)
IgG	17 (15.2)	18 (22.0)	0.228 (0.64; 0.31–1.33)
IgM and/or IgG	36 (32.1)	31 (37.8)	0.413 (0.78; 0.43–1.42)
aβ2GP1 N (%)			
IgM	12 (10.7)	19 (23.2)	**0.031 (0.42; 0.19–0.92)**
IgG	27 (24.1)	32 (39)	**0.033 (0.51; 0.27–0.95)**
IgM and/or IgG	33 (29.5)	35 (42.7)	0.070 (0.58; 0.32–1.05)
Only LA N (%)	50 (44.6)	24 (29.3)	**0.030 (1.95; 1.07–3.57)**
Only aCL N (%)	5 (4.5)	2 (2.4)	0.462 (1.87; 0.35–9.88)
Only aβ2GP1 N (%)	3 (2.7)	4 (4.9)	0.424 (1.86; 0.41–8.56)
N° of aPL			
1 N (%)	57 (50.9)	28 (34.1)	**0.021 (1.99; 1.11–3.59)**
2 N (%)	16 (14.3)	13 (15.9)	0.782 (1.12; 0.51–2.48)
3 (triple positive) N (%)	21 (18.8)	23 (28.0)	0.137 (1.67; 0.85–3.29)
Thrombotic aPL profile			
Low thrombotic risk N (%)	24 (21.4)	23 (28.0)	0.289 (1.43; 0.74–2.77)
High thrombotic risk N (%)	88 (78.6)	59 (72.0)	0.289 (1.43; 0.74–2.77)

Abbreviations: Antiphospholipid autoantibodies (aPL); Anti β2 glycoprotein 1 (aβ2GP1); Anti cardiolipin (aCL); Lupus anticoagulant (LA).

**Table 3 ijms-24-16270-t003:** Coagulation FXI levels in patients with aPL, with (APS) or without (AaPL) vascular complications.

	APS (n:112)	AaPL (n:82)	Univariate Analysis *p* (OR; 95%CI)
FXI:C			
SFX Mean ± SD (%)	122.8 ± 33.4	104.5 ± 27.5	**<0.001**
SS Median (IQR) (%)	110.9 (46.9)	99.1 (36.2)	**0.002**
FXI:C > 150%			
SFX N (%)	22 (19.6)	2 (2.4)	**0.002 (10.11; 2.30–44.36)**
SS N (%)	20 (17.9)	3 (3.7)	**0.005 (5.91; 1.69–20.65)**
Global N (%)	17 (15.2)	1 (1.2)	**0.009 (14.95; 1.95–114.83)**
FXI:C < 70%			
SFX N (%)	3 (2.7)	10 (12.2)	**0.018 (4.93; 1.31–18.54)**
SS N (%)	10 (8.9)	16 (19.5)	**0.042 (2.41; 1.03–5.64)**
Global N (%)	3 (2.7)	10 (12.2)	**0.018 (4.93; 1.31–18.54)**

**Table 4 ijms-24-16270-t004:** Multivariate analysis of risk factors for APS in patients with aPL.

	B	Standard Error	OR	CI95%	*p*
Dyslipidemia	0.770	0.334	2.16	1.12–4.16	0.021
LA	0.797	0.355	2.22	1.12–4.45	0.025
aβ2GP1 IgG	−0.966	0.353	0.38	0.19–0.76	0.006
FXI:C > 150%	2.448	1.052	11.57	1.47–90.96	0.020
FXI:C < 70%	−1.738	0.347	0.18	0.36–0.86	0.032

anti β2 glycoprotein B1 (aβ2GP1); Lupus anticoagulant (LA).

## Data Availability

All data is available upon request via email to the corresponding author. MEMB has full access to all data in the study and takes responsibility for its integrity and data analysis. To obtain the original data, please contact uge2985@hotmail.com.

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
