# Peer review of "Factor XI in Carriers of Antiphospholipid Antibodies: Elevated Levels Associated with Symptomatic Thrombotic Cases, While Low Levels Linked to Asymptomatic Cases"

_ijms, 2023, doi:10.3390/ijms242216270_

Round 1

Reviewer 1 Report

Comments and Suggestions for Authors

In their manuscript „Factor 11 in carriers of antiphospholipid antibodies: elevated levels associated with symptomatic thrombotic cases, while low levels linked to asymptomatic cases.“ Pagán-Escribano et al have been investigated factor 11 concentrations in correlation to antiphospholipid-antibodies. In a second approach the authors tried to establish core mutations in cases were factor 11 expression was under expressed.

The topic is of clinical relevance and the results primarily appear to be of importance to further explanation of thrombotic events in APS. In order to ensure the validity of this report, there are some major and minor points that need to be addressed prior to publication. I am fully aware of the complicated nature of a study in single-center setting and the fact that clinical presentation of APS remains a rather rare disease. I still hope that the authors may address the following points, at least to some extent, as this may render their work an excellent paper and reflect the amount of work and diligence it took to put this comprehensive dataset together.

Major: 

Table 1 and Table 2: Readability needs to be improved. Essential results are buried under unnecessary data (e.g. risk factors for vtem thrombophlia screening).

Figure 1 and Figure 2: Both figures are well scripted but included data are too similar, unsing just one figure instead should be sufficient.

While the authors can detect a significant level of reduction in FXI:C < 70% in AaPL this can be seen in just up to 8.9% of symptomatic antiphospholipid syndrome patients. It remains unclear why healthy individuals show no underexpression of factor XI:c in total. Question arises when low level of Factor XI:C are detected in APS but none in healthy individuals if the right target for the thesis had been selected. The authors state that a healty blood donor, presumably already integrated in healthy individuals in Figure 2 B, carried a mutation that was identified in 2 patients with FXI deficiency. Therefore Figure 2B needs at least some critical contention as if the displayed Factor XI:C is really displayed properly.

Figure 3 reveals the most critical weakness. Of the 16 detected FXI:C deficient individuals only 4 could be verified in plasma antigen by western blot. All of them showed genetic variations. Of these 4 genetic variations one can be found in one asymptomatic (AaPL) as well as one symptomatic (APS). Of the two remaining carried mutations leading to underexpression of FXI, one can be found in an healthy individual and the last one in AaPL. Therefore all detected mutations are distributed 1:2:1. It seems questionable whether the mutations detected here alone can serve as a sufficient explanation for the Factor XI reductions.

Minor:

Some minor spelling mistakes should be improved.

Comments on the Quality of English Language

Some editing should be applied

Author Response

Reviewer 1

In their manuscript „Factor 11 in carriers of antiphospholipid antibodies: elevated levels associated with symptomatic thrombotic cases, while low levels linked to asymptomatic cases.“ Pagán-Escribano et al have been investigated factor 11 concentrations in correlation to antiphospholipid-antibodies. In a second approach the authors tried to establish core mutations in cases were factor 11 expression was under expressed.

The topic is of clinical relevance and the results primarily appear to be of importance to further explanation of thrombotic events in APS. In order to ensure the validity of this report, there are some major and minor points that need to be addressed prior to publication. I am fully aware of the complicated nature of a study in single-center setting and the fact that clinical presentation of APS remains a rather rare disease. I still hope that the authors may address the following points, at least to some extent, as this may render their work an excellent paper and reflect the amount of work and diligence it took to put this comprehensive dataset together.

Dear Reviewer 1, we sincerely appreciate your thoughtful comments and feedback on our manuscript.

Major: 

 Table 1 and Table 2: Readability needs to be improved. Essential results are buried under unnecessary data (e.g. risk factors for vtem thrombophlia screening).

We appreciate your feedback and agree that the tables should be easier to read. We have simplified the data to make it more comprehensible for readers.

Figure 1 and Figure 2: Both figures are well scripted but included data are too similar, unsing just one figure instead should be sufficient.

Thank you for your comment. In response to your suggestion, we have include the graphs into the same figure but presented them in different panels. We believe that both approaches provide complementary insights into the data, which enhances the robustness of our results. Panel A displays the median FXI levels for each group, while Panels B and C compare the distribution of patients falling within specific value ranges (high: FXI:C>150% or low: FXI:C<70%) across the different groups, according to their clinical impact: risk of or protection against thrombosis, respectively.

While the authors can detect a significant level of reduction in FXI:C < 70% in AaPL this can be seen in just up to 8.9% of symptomatic antiphospholipid syndrome patients. It remains unclear why healthy individuals show no underexpression of factor XI:c in total. Question arises when low level of Factor XI:C are detected in APS but none in healthy individuals if the right target for the thesis had been selected. The authors state that a healty blood donor, presumably already integrated in healthy individuals in Figure 2 B, carried a mutation that was identified in 2 patients with FXI deficiency. Therefore Figure 2B needs at least some critical contention as if the displayed Factor XI:C is really displayed properly.

We apologize for the oversight in presenting our results. As expected, healthy subjects do not exhibit reduced levels of FXI. FXI deficiency is a rare condition in Caucasians, although it is believed to be underdiagnosed by various research groups. In our series of healthy controls, we identified only one out of 222 subjects with FXI levels below 70%. The data represented in the violin plot have and overall mean of 99.6±19.0% by SFX; and 101.1±17.5% by SS, including the case with congenital FXI deficiency (Figure 1A in the new version of the manuscript). Upon sequencing the F11 gene, we discovered that this control carried the c.1327C>T, p.Arg443Cys mutation in heterozygous state. This mutation has been previously associated with congenital FXI deficiency at the HGMD (CM062624). The coagulant FXI levels in this subject were 23% by SFX and 24% by SS. This reduction is attributed to the presence of this heterozygous variant. Accordingly, in the Figure 1C that represents the percentage of cases with FXI:C levels below 70%, only one is counted for healthy controls (1/222, 0.4%).

On the other hand, the almost total absence of subjects with FXI >150% in the healthy control group (only 2/222, with 154% and 151% values) represented in figure 1B (previously 2A) was also expected, as the range of normality of FXI:C is 70-150% in healthy population, and levels > 150% have been described in some disorders, particularly in thromboembolic disorders.

Figure 3 reveals the most critical weakness. Of the 16 detected FXI:C deficient individuals only 4 could be verified in plasma antigen by western blot. All of them showed genetic variations. Of these 4 genetic variations one can be found in one asymptomatic (AaPL) as well as one symptomatic (APS). Of the two remaining carried mutations leading to underexpression of FXI, one can be found in an healthy individual and the last one in AaPL. Therefore all detected mutations are distributed 1:2:1. It seems questionable whether the mutations detected here alone can serve as a sufficient explanation for the Factor XI reductions.

The reviewer correctly points out that the mutations detected may not be the sole explanation for the reduction in Factor XI levels. Indeed, the presence of a pathogenic mutation in F11 was only detected in cases with antigen deficiency according to Western blot results (CRM- deficiency). However, all cases with FXI deficiency with no reduction of FXI antigenic levels and no F11 mutation had antiphospholipid antibodies (aPL), most of them AaPL (N= 8) but also 2 APS patients. Based on previously reported data supporting an inhibitory activity of aPL on FXI, we speculated, that certain aPL might act as inhibitors of FXI, explaining the FXI deficiency found by coagulometric assays. We confirmed this hypothesis and the presence of a FXI inhibitor by dilution assays. Interestingly, and consistent with previous studies, we found an inversely proportional correlation between aβ2GP1 levels and FXI coagulation activity (Shi, T.; et al. Β2-Glycoprotein I Binds Factor XI and Inhibits Its Activation by Thrombin and Factor XIIa: Loss of Inhibition by Clipped Β2-Glycoprotein I. P Natl Acad Sci USA 2004, 101, 3939–3944, doi:10.1073/pnas.0400281101.   Rahgozar, S.; et al. Beta2-Glycoprotein I Binds Thrombin via Exosite I and Exosite II: Anti-Β2-Glycoprotein I Antibodies Potentiate the Inhibitory Effect of Β2-Glycoprotein I on Thrombin-Mediated Factor XIa Generation. Arthritis Rheum. 2007, 56, 605–613, doi:10.1002/art.22367). The overrepresentation of AaPL vs APS in this group of patients was explained by the protective role of reduced FXI activity. These statements have been included in the revised version of our manuscript.

Conversely, low FXI levels (< 70%) may protect against thrombosis, as low FXI levels were 4 times more common in patients with positive aPL than in those who developed APS. Although cases of congenital FXI deficiency have been identified, in our cohort FXI deficiency was mainly acquired, probably caused by aPL itself. Thus, neutralization of FXI by aPL may reduce or abolish its prothrombotic consequences and protect against a thrombotic event, mimicking the effect of therapeutic anti-FXI antibodies. Consistent with previous studies, we found an inversely proportional correlation between aβ2GP1 levels and FXI coagulation activity [4,5]. The overrepresentation of AaPL vs APS in this group of patients might be explained by the protective role of reduced FXI activity.

We would emphasize, based on our study that the reduced FXI levels, whether due to genetic factors or acquired factors (such as antibodies or inhibitors), could potentially serve as a protective factor against thrombosis. In our study, FXI:C <70 % was more frequent in AaPL subjects than in thrombotic patients (APS or THROMB groups) or healthy subjects: 10 AaPL; 3 APS; 2 THROMB, and 1 HC. Congenital FXI deficiency was confirmed in 2 AaPL cases, 1 APS, and 1 Healthy control.

To enhance the clarity of Figure 2, we have simplified the information and revised the names of each sample.

Minor:

Some minor spelling mistakes should be improved.

We have revised the spelling through the manuscript.

The manuscript has undergone English revision, the references have been adjusted to match the journal's style, and the material and method, and conclusion sections have been organized to meet the journal's requirements.

Reviewer 2 Report

Comments and Suggestions for Authors

It is a well designed study regarding a topic with a great interest in internal medicine.  It is well written and the text is clear and easy to read. In introduction it is analysed with a sufficient way the theory of the study. Methods and materials have introduced the single centre collected data and the methods of molecular analysis. It is really a small number of samples from one single centre and the number of patients not really so big to make a clear conclusion. It is a new hypothesis in APS regarding the levels of FXI and the risk of thrombosis so the sample could be larger.

Even though there are no similar studies, only one, to strongly recommend that factor XI may play a role ti thrombotic episodes in APL. But I have to point out that the statistical analysis is good and the conclusions consistent with the arguments that are presented. In discussion references are mentioned and only one study is similar to the present one. The literature may need this study to be published to prove or not the hypothesis that the authors are doing.

The results demonstrate the importance of the recommendation of the authors. Tables and figures need to be more clarified.

Comments on the Quality of English Language

Moderate changes 

Author Response

Reviewer 2

It is a well designed study regarding a topic with a great interest in internal medicine.  It is well written and the text is clear and easy to read. In introduction it is analysed with a sufficient way the theory of the study. Methods and materials have introduced the single centre collected data and the methods of molecular analysis. It is really a small number of samples from one single centre and the number of patients not really so big to make a clear conclusion. It is a new hypothesis in APS regarding the levels of FXI and the risk of thrombosis so the sample could be larger.

Even though there are no similar studies, only one, to strongly recommend that factor XI may play a role ti thrombotic episodes in APL. But I have to point out that the statistical analysis is good and the conclusions consistent with the arguments that are presented. In discussion references are mentioned and only one study is similar to the present one. The literature may need this study to be published to prove or not the hypothesis that the authors are doing.

The results demonstrate the importance of the recommendation of the authors. Tables and figures need to be more clarified.

Thank you for the positive comments of reviewer 2 on our manuscript.

We have made efforts to improve the clarity of the tables and figures in the revised version of the manuscript.

Round 2

Reviewer 1 Report

Comments and Suggestions for Authors

Improvements have been supplied sufficiently. I suggest accepting the manuscript in the current form.